# Investigating How Genetic Merit and Country of Origin Impact the Profitability of Grass-Based Sheep Production Systems

**DOI:** 10.3390/ani13182908

**Published:** 2023-09-13

**Authors:** Nicola Fetherstone, Fiona M. McGovern, Noirin McHugh, Tommy M. Boland, Alan Bohan

**Affiliations:** 1Animal and Grassland Research and Innovation Centre, Teagasc, Mellows Campus, Athenry, Co., Galway H65 R718, Ireland; 2School of Agricultural Science, University College Dublin, Dublin D04 V1W8, Ireland; 3Animal and Grassland Research and Innovation Centre, Teagasc, Moorepark, Fermoy, Co., Cork P61 P302, Ireland; 4Teagasc Advisory and Education, Ballymote, Carrownanty, Co., Sligo F56 A585, Ireland

**Keywords:** genetic indexes, profitability, sheep, bio-economic model

## Abstract

**Simple Summary:**

An animal’s genetic merit can govern its production potential and performance. This study used a bio-economic model informed by a large dataset of records from a commercial sheep-production-flock study to investigate the effects of selecting breeding females based on their maternal genetic merit for productivity, feed demand and gross margin. The results showed that selecting and using breeding females of high genetic merit will increase the gross margin of the flock, thus improving the profitability of the production system for the farmer. Our results provide insight for sheep-meat-production industries into the cumulative economic returns of using animals of high genetic merit. Our results also indicate potential outcomes from improving flock genetic merit, an important key performance indicator for the industry, and inform decision-making around production priorities of producers.

**Abstract:**

The objective of this study was to simulate and assess the profitability of sheep production systems that varied in maternal genetic merit (high or low) and country of origin (New Zealand (NZ) or Ireland), using the Teagasc Lamb Production Model (TLPM). A production system study performed at Teagasc Athenry, Co. Galway, Ireland, from 2016 to 2019, inclusive, provided key animal performance input parameters, which were compared across three scenarios: high maternal genetic merit NZ (NZ), high maternal genetic merit Irish (High Irish) and low maternal genetic merit Irish (Low Irish). Prior to the beginning of the study ewes and rams were imported from New Zealand to Ireland in order to compare animals within the same management system. Ewes were selected based on the respective national maternal genetic indexes; i.e., either the New Zealand Maternal Worth (NZ group) or the €uro-star Replacement index (Irish groups). The TLPM was designed to simulate the impact of changes in physical and technical outputs (such as number of lambs, drafting rates and replacement rates) on a range of economic parameters including variable costs, fixed costs, gross margin and net profit. Results showed that total farm costs (variable and fixed) were similar across the three scenarios, driven by the similar number of ewes in each scenario. The number of lambs produced and the cost of production per lamb was 14.05 lambs per hectare for the NZ scenario at a cost of EUR 82.35 per lamb, 11.40 lambs per hectare for the High Irish scenario at a cost of EUR 101.42 per lamb and 11.00 lambs per hectare for the Low Irish scenario at a cost of EUR 105.72 per lamb. The net profit of the three scenarios was EUR 514, EUR 299, and EUR 258 per hectare, for the NZ, High Irish and Low Irish scenarios, respectively. Overall, the NZ scenario had a lower cost of production in comparison to either Irish group, while the High Irish scenario had a 14% greater net profit than the Low Irish scenario, equating to an additional EUR 41 per hectare net profit. Output from this simulation model reiterates the importance, for overall farm profitability, of maximising the number of lambs weaned per hectare, particularly through maximising income and diluting the total farm costs. To conclude, the use of high-maternal-genetic-merit animals, regardless of their country of origin impacts farm profitability.

## 1. Introduction

Farm production gains can be accelerated through the selection of animals using genetic indexes, as shown previously in beef [1], dairy [2] and sheep [3] farms. The validation of genetic indexes is essential in order to increase farmers’ confidence in breeding programmes and to demonstrate that genetic progress can be achieved through the use of animals of superior genetic merit. Previous research on the impact of maternal genetic merit and country of origin on the productivity of sheep production systems has shown that differences exist in reproduction and lambing performance [4], lamb growth and performance [5], and ewe performance and efficiency [6]. Although differences were often biologically small between the three genetic-merit groups within the previously reported literature, the overall financial impact of the accumulation of these differences at the full farm-system level has not been investigated to date. The economic assessment of dairy cows divergent on genetic merit has been reported previously [7], and demonstrated that selecting high-genetic-merit dairy cows resulted in an increase in both farm productivity and profitability; however, to the authors knowledge, no such controlled experiment has been undertaken on sheep divergent in genetic merit. 

Bio-economic models are used to simulate changes in production, price or policy parameters, so that their impact can be quantified prior to the implementation of such changes on-farm [8]. The Teagasc Lamb Production Model (TLPM), a bio-economic model used to simulate Irish sheep production systems [9], has previously been used to quantify the impact of stocking rate and prolificacy potential of a ewe flock [10]. Others have performed similar economic analysis in order to assess the profitability of dairy systems of various genetic-merit groups [7] and calving patterns [11], of dairy to beef systems for various breeds and finishing ages [12], of suckler cow systems for varied herd size, composition and diet [13], and of pig systems for various nutrition plans and accommodation facilities [14].

The objective of this study, therefore, was to use previously published farm system data [4,5,6] to quantify the profitability of a sheep production system varying in maternal genetic merit and country of origin, using the TLPM. In order to examine the robustness of the Irish national genetic indexes it was decided to compare animals of New Zealand origin, selected from a genetic index based on similar breeding goals, to animals of Irish origin in a common environment.

## 2. Materials and Methods

### 2.1. Bio-Economic Model

The TLPM is a whole-farm bio-economic simulation model, built in an excel spreadsheet, and used to simulate sheep production systems in Ireland [9]. The model can simulate the impact of institutional, technical or environmental changes that may vary within a production system and quantify the consequence of those changes on both physical and technical outputs including variable costs, fixed costs and net profit [9]. The livestock inventory in the model includes a breakdown of: lambs, replacement ewe lambs, replacement hoggets, mature ewes and rams. The livestock inventory then enables the model to generate a monthly feed budget, based on flock net energy requirements [15,16] comprising grass, grass silage and concentrate, depending on time of year and stage of production. The TLPM simulates a full production year commencing at mating, and incorporates feed, land and labour requirements, as well as drafting patterns linked to the animals within the inventory. Outputs generated from the TLPM model include: cash-flow forecasts, profit, loss and balance sheet, feed budgets, livestock trading inventories and physical ratios. The estimated farm profit is presented on a total farm profit, per hectare and/or per ewe joined basis. 

### 2.2. Maternal-Genetic-Merit and Country-of-Origin Scenarios

The three scenarios investigated in the present study we used data obtained from a production system study that was performed over a four-year period (2016 to 2019, inclusive) at Teagasc Athenry. All ewe reproductive and lambing [4], lamb growth and performance [5], and ewe efficiency and performance [6] data were sourced from the performance recording of this flock, better known as the INZAC flock. The study consisted of two maternal-genetic-merit types: high or low genetic merit, and two countries of origin: New Zealand (NZ) and Ireland. Economic indices determining animals of both high or low genetic merit had been developed previously [17] and validated [18] using field data from the Irish national flock. The study design and flock management has been reported in detail elsewhere [4]. Briefly, using New Zealand Maternal Worth, NZ ewes were ranked as the top 40%, and had been imported into Ireland in 2013 and 2014, while Irish ewes were selected based on their maternal genetic merit and classified as the top or bottom 20% within breed on the €uro-star Replacement index [4,19], at the time of entry to the study. For the purposes of the TLPM, each scenario was simulated on a 20-hectare farmlet consisting of 240 purebred Texel (n = 120) and Suffolk (n = 120) ewes, stocked at 12 ewes per hectare. The key performance indicators for each scenario, describing reproductive performance and lambing [4], lamb growth and performance [6], and ewe efficiency and performance [5] were retrieved and inputted to the TLPM, as outlined in Table 1 and Table 2. Selection indices for this study were based on those used in the Irish national breeding objectives [1] and hosted by Sheep Ireland (https://www.sheep.ie; accessed on 10 January 2021). 

#### 2.2.1. Scenario 1: New Zealand Ewes of High Genetic Merit

In the first scenario (NZ scenario), ewes and rams of NZ origin that had been imported to Ireland were used. The 240 ewes of NZ origin were mated by rams of the same breed and genetic-merit group. The average pregnancy scan rate of the NZ scenario was 1.84 lambs per ewe joined to the ram and a total of 224 ewes lambed down when barren ewes (7.23%) were removed. A total of 350 lambs were weaned in the NZ scenario, which equates to 1.46 lambs weaned per ewe joined to the ram. An additional 52 lambs were sold off the ewe within the first week of life for artificial rearing. The average weaning weight of lambs reared on ewes was 33.99 kg. Drafting started in June, which coincided with weaning, and 94% of lambs were drafted by 1 October from a grass-only diet. The average carcass weight achieved was 20.04 kg. Ewe replacement rate was 29%, which included the culling of 56 ewes, and 13 ewes died during the production year. The average annual grass growth rate was 10,792 kg dry matter (DM) per hectare, of which 9180 kg DM per hectare was utilised (85%).

#### 2.2.2. Scenario 2: Irish Ewes of High Genetic Merit

In the High Irish scenario, the average pregnancy scan rate was 1.62 lambs per ewe joined to the ram and 223 ewes lambed down once barren ewes (7.23%) were removed. A total of 308 lambs were weaned in the High Irish scenario, equating to 1.28 lambs weaned per ewe joined to the ram. An additional 46 lambs were sold for artificial rearing. The average weaning weight of lambs reared on ewes was 33.14 kg. Drafting started in June, which coincided with weaning, and 88% of lambs were drafted by 1 October from a grass-only diet. The average carcass weight achieved was 20.12 kg. Ewe replacement rate was 34%, of which 69 ewes were culled and 13 ewes died during the production year. The annual grass growth rate averaged 10,792 kg DM per hectare of which 85% was utilised.

#### 2.2.3. Scenario 3: Irish Ewes of Low Genetic Merit

In the Low Irish scenario the average pregnancy scan rate was 1.59 per ewe joined to the ram and a total of 222 ewes lambed down once barren ewes (7.23%) were removed. A total of 303 lambs were weaned in the Low Irish scenario, which equates to 1.26 lambs weaned per ewe joined to the ram. Average weaning weight of lambs reared on ewes was 32.37 kg. Drafting commenced at weaning, and 84% of lambs were drafted by 1 October from a grass-only diet. The average carcass weight achieved was 20.22 kg. Ewe replacement rate was 36%; a total of 72 ewes were culled and 13 ewes died during the production year. The annual grass growth rate averaged 10,792 kg DM per hectare, of which 9180 kg DM per hectare was utilised.

### 2.3. Model Assumptions

Ewe lambs were retained from within the flock and mated for the first time at approximately 19 months of age and lambed down for the first time at two years of age. Ewe mature weight (at mating) and barren rates averaged 81.19 kg and 7.23%, respectively, across the three groups. All groups were housed for the winter period, commencing on the 20 December and returned to grass within 48 h of lambing, commencing 1 March. Concentrate feeding was offered to all ewes pre-lambing, according to their litter size but ceased on return to grass post-lambing. The mean lambing date was 8 March. Lamb birth weight averaged 5.16 kg and lamb mortality averaged 15.9% from birth to drafting. Surplus lambs that were sold off for artificial rearing were sold within the first week of life in order to represent a production system with a maximum rearing type of two lambs per ewe at pasture. Lambs were selected for drafting (grass-only diet) once the target live weight was reached; the targeted drafting weight was 43 kg in June, and increased by a kilogram per month thereafter to account for the reduction in carcass kill-out as the season progressed. Across all scenarios, concentrate supplementation was offered to all lambs, that had not been drafted by the 1 October, at a rate of 0.350 kg per day, until they reached the targeted drafting weight. Fertiliser application was the same across each scenario with nitrogen, phosphorus and potassium applied at a rate of 130 kg nitrogen per hectare, 16 kg phosphorus per hectare and 33 kg potassium per hectare across the production year. 

### 2.4. Economic Assumptions

Labour requirement was assumed at eight hours per ewe [20], with the main labour unit working up to a maximum of 300 h per month. Any additional requirement was fulfilled by hired labour at a rate of EUR 10 per hour. Each lamb sold for artificial rearing within the first week of life was valued at EUR 25. Variable and fixed costs were assigned based on the 5-year average industry prices [9].

### 2.5. Sensitivity Analysis

To investigate the impact of fluctuations in input parameters on each of the three modelled scenarios, sensitivity analysis was conducted on two stochastic input variables, namely lamb price and concentrate price, where the prices of each variable fluctuated by ±10% and their impact on farm profitability was quantified. 

## 3. Results

### 3.1. Physical

Physical performance parameters from this study have been presented in detail [4,5,6] for reproduction and lambing, lamb growth and performance, and ewe performance and efficiency; however, to summarise, the weaning rate ranged from 1.26 (Low Irish) to 1.46 (NZ) lambs per ewe joined (Table 1). The number of lambs weaned per hectare was 17.5, 15.4 and 15.2 lambs for the NZ, High Irish and Low Irish scenarios, respectively. The quantity of carcass weight produced for the NZ, High Irish and Low Irish was 282, 230 and 222 kg per hectare, respectively (Table 2). 

A greater proportion of the Low Irish lambs (16%) were not drafted by October, in comparison to the NZ and High Irish scenarios (6.0 and 11.6%, respectively). The total concentrates (ewes and lambs) fed in each scenario equated to 5466 (NZ) kg, 5199 (High Irish) kg and 5260 (Low Irish) kg, of which 460 (NZ) kg, 748 (High Irish) kg and 883 (Low Irish) kg was fed to lambs. The silage requirements totalled 20,858 (NZ), 21,909 (High Irish) and 22,094 (Low Irish) kg DM.

### 3.2. Financial

Total farm income ranged from EUR 1419 (Low Irish) to EUR 1672 (NZ) per hectare, with lamb sales accounting for a large proportion of the total income and ranging from EUR 1124 (Low Irish) to EUR 1437 (NZ) per hectare. Total farm costs were similar across the three scenarios (Table 3).

The main contributors towards the variable costs were fertiliser, veterinary/medicine, machinery operation and repair, and silage production costs, which, on average, accounted for EUR 572 (66.2%) of the total average variable costs of EUR 864. The cost of producing a lamb for slaughter varied considerably between the scenarios and ranged from EUR 82.35 (NZ) and EUR 105.72 (Low Irish; Table 4). 

The net profit per lamb slaughtered varied from EUR 23.44 (Low Irish) to EUR 36.54 (NZ). As the number of lambs slaughtered within each scenario increased, the net profit produced per hectare also increased; the NZ scenario produced 14.05 lambs per hectare and generated a net profit of EUR 514 per hectare, the High Irish scenario produced 11.40 lambs per hectare and generated a net profit of EUR 299 per hectare, and the Low Irish scenario produced 11.00 lambs per hectare and generated a net profit of EUR 258 per hectare. The net profit reported on a per-ewe basis was EUR 21, EUR 25 and EUR 43 per ewe for Low Irish, High Irish and NZ groups, respectively. Despite the similarity of total farm costs across the three scenarios, the profitability of the NZ scenario, whether reported as total farm net profit, on a per hectare, per lamb or per ewe basis, was always the greatest, followed by the High Irish and then the Low Irish, as dictated by the number of lambs sold from the system annually. 

### 3.3. Sensitivity Analysis

The impact of fluctuations in lamb and concentrate prices (±10%) on total farm net profit for each scenario is presented in Table 5. 

The total net profit, regardless of fluctuation in lamb or concentrate prices, was always greatest for the NZ scenario, followed by High Irish and lowest for the Low Irish scenario. A ±10% fluctuation in concentrate prices resulted in the total net profit changing by ±1.6%, ±2.6% and ±3.0% across the NZ, High Irish and Low Irish scenarios, respectively. Fluctuating lamb prices had a greater impact on the total net profit across all three scenarios in comparison to fluctuating concentrate prices, whereby the total net profit changed by ±26.7%, ±39.3% and ±41.5% across the NZ, High Irish and Low Irish scenarios, respectively.

## 4. Discussion

Quantifying the financial impact of selecting breeding animals based on their maternal genetic merit was evaluated in the current study. A range of production data including parameters that are used within the national genetic evaluations in Ireland, such as: lamb survival, ewe mature weight, days to slaughter and number of lambs born, have been reported [4,5,6], where performance recording was conducted as part of a production-system study. This allowed the replication of the actual on-farm performance of the experimental groups: NZ, High Irish and Low Irish, to be simulated using the TLPM. The concept of using actual on-farm data as the input parameters was used [10] when comparing the impact of different stocking rates and prolificacy potential on the profitability of sheep production systems [21]. Other studies across dairy [22,23,24], beef [25] and sheep [3,26,27] farms have previously compared the performance of animals of various genetic-merit groups; however, to the authors knowledge few have then used controlled studies to simulate an economic assessment of such systems [10] using data supplied [21]. 

### 4.1. Physical

Number of lambs born, weaned or sold has been shown to be one of the most important determinants of profitability in meat-sheep systems of production [10,28]. Number of lambs born is often included as a goal trait within national sheep-breeding objectives in Slovakia [29] and Canada [30], with a large emphasis placed on both the number of lambs born and lamb survival within both the Irish and New Zealand maternal indexes [17,19]. Results showed that an increase in the number of lambs born and weaned can be achieved through the use of high-genetic-merit animals, regardless of country of origin [4]. Results from this study corroborate previous research that has demonstrated the potential to improve the economic performance of lamb production systems through an increase in the number of lambs produced [28]. As the proportions of barren ewes, lamb mortality and lambs sold for artificial rearing were the same across each scenario, the same values were inputted for each scenario. This showed that higher pregnancy scan rate was the main driver of performance in achieving a greater number of lambs weaned and/or slaughtered per ewe. However, it is noteworthy that a higher litter size also had higher concentrate costs, i.e., the NZ scenario (Table 3), albeit costs were diluted across the greater proportion of lambs slaughtered.

Replacement rates across all three scenarios were higher (28.80%, 34.14% and 35.59% for NZ, High Irish and Low Irish, respectively) than those previously reported by a cohort of Irish farmers surveyed [31], where replacement rates averaged 22.4%. This was the result of a strict culling regime where reasons for culling included barrenness for two consecutive years, prolapse, or mastitis. This was implemented in practice on-farm, and using purebred animals for which replacement rates has not been widely reported in the past. Furthermore, such reasons for culling are not widely reported at a commercial level for individual ewes, resulting in ewes surviving longer in the flock and a reduced replacement rate. The additional costs associated with retaining and managing a high number of replacements per year has been discussed elsewhere [32], and is known to impact variable costs within the system, including silage production and veterinary costs (Table 3). A quantity of surplus silage bales was sold from the High and Low Irish systems as a result of surplus grass production, however, it must be considered whether utilisation of this grass in supporting an increased weaning rate per ewe joined may have been a more effective use of this resource.

Due to constraints of the experiment being carried out across all three scenarios in this study, a large proportion of surplus lambs were removed from the system when sold for artificial rearing (11.8%). Although sold at a flat rate of EUR 25 per lamb within the first week of life, the total farm income could potentially be further increased through the artificial rearing of these lambs on farm, allowing total costs to be divided over a greater number of lambs. Previous research has demonstrated that through careful management practices, there is potential to achieve a similar cost of production per lamb for those artificially reared, as for those reared by a ewe [33]. Therefore, the rearing of lambs artificially or reared as triplets by the ewe should be considered. However, while the rearing of triplet lambs would increase the overall output per hectare [10], the individual output per animal would decline through a reduction in lamb growth, drafting rates and lamb survival [34]. In a recent survey, only 25% of Irish sheep farmers reported that lambs born as triplets, were reared as triplets. Therefore, to imitate current on-farm management practices, no lambs were reared as triplets within the current study; instead, lambs were either cross-fostered onto other ewes (where possible) or sold within the first week of life.

Growing and utilising the target quantity of grass required in order to meet flock demand is key when attempting to keep the costs incurred as a result of concentrate usage within the diet low. Previous research has demonstrated that concentrate costing in the region of EUR 260 to EUR 350 per tonne, was 300 to 690% more expensive than grazed grass [35]. Therefore, as a large proportion of the animal’s diet within this study consisted of grazed grass and silage (~98%), low concentrate costs were associated with each scenario. This demonstrates the importance of achieving optimum lamb growth and drafting rates early in the season so that the number of lambs remaining within the system as grass quality diminishes and growth declines (approximately 1 October) is minimised [36]. This study highlights the importance of maximising grazed grass in the diet of the animal and is most reproducible in regions where an extended grazing season is permitted. 

### 4.2. Financial

As expected, increasing lamb output per hectare and reducing the cost of production resulted in an increased net profit per hectare. The difference in the cost of production per lamb between scenarios can be attributed to the increased costs involved in maintaining more ewes per lamb slaughtered. Therefore, findings from this study agree with previous research that the number of lambs weaned per hectare is critical to the profitability of the production system [10,28]. The number of lambs weaned per hectare was similar to the 11.5 to 14.4 lambs per hectare range reported in [37].

The Irish National Farm Survey results indicate that sheep farms weaning 12 lambs per hectare were within the top third of farms in terms of profitability and generated EUR 912 gross margin per hectare [38], which is greater than achieved across any of the three scenarios in this study at EUR 809, EUR 595 and EUR 554 per hectare for NZ, High Irish and Low Irish, respectively. The exclusion of any subsidies or payments within the TLPM contributes somewhat to this difference, however an average farm received EUR 60 per hectare in payments each year [39]. When compared to the performance of mid-season lamb producers [39], areas for potential improvement can be identified. Fundamental to improving this performance is the reduction of single rearing ewes (if artificially reared lambs are to be continued to be removed from the system) or else the rearing of triplet lambs by their dams in order to increase carcass output per hectare, and dilute the variable and fixed costs across a greater number of lambs [21]. Furthermore, although the overall lamb mortality rates, i.e., birth to drafting, reported in the present study (15.9%) were similar to those previously reported (11.80%) for purebred animals [40], the rates were greater than the Teagasc 2025 Sheep road map target of ≤8% set for mid-season lamb producers, whereby 61% were achieving this target in 2019 [39].

Superior performance in terms of the number of lambs slaughtered per hectare (16.75 lambs per hectare) and a lower cost of production per lamb (EUR 75.69 per lamb) was observed previously in comparison to this study [21]. Eighty-two percent of the income generated from the carcass value was required to break even given the cost of production, while the remainder could be attributed to increasing the net profit of the system [10]. However, within the current study, 80.6%, 98.6% and 103.3% of the income generated from the carcass value was required to cover the cost of production, meaning that the other income sources, such as wool, silage, or income generated from sales of cull ewes, generated a greater proportion of the net farm profit for the Low Irish system in order to allow them to generate a net profit per lamb, rather than a net loss. As the Irish €uro-star genetic indexes report the ranking of the animal based on the additional profit generated per lamb born in comparison to the national average, it is also important to compare the net profit per lamb born within each scenario, i.e., EUR 36.54, EUR 26.19 and EUR 23.44 per lamb born for NZ, High Irish and Low Irish, respectively. Albeit the net profit reported is not relative to the national average as in the index, the Low Irish scenario was still the least profitable. The NZ scenario unexpectedly outperformed the High Irish animals given the average €uro-star Replacement index values at the start of the study of EUR 0.06 ± 0.741, EUR 1.04 ± 0.617 and EUR 0.68 ± 0.729 per lamb born for the NZ, High Irish and Low Irish scenarios, respectively. However, the low level of accuracy reported for the NZ animals at the start of the study (34%) was noteworthy, due to very little data recorded prior to their importation, in comparison to the High Irish (44%) and Low Irish (42%) animals. Therefore, differences in profitability of the three scenarios were not as expected, especially between the High Irish and Low Irish animals where a 14% difference in net profit was observed. It is also noteworthy that the profitability of Ireland’s sheep industry is likely to increase further through the selection of foreign maternal breeds ranking higher on the maternal index, in comparison to terminal breeds (i.e., Texel and Suffolk ewes) which formed the production data in the present study. However, this would require a major shift in the breeding structure of the Irish sheep industry, hence the decision to select foreign animals from some of the most commonly used breeds in Ireland.

### 4.3. Sensitivity Analysis

Sensitivity analysis was undertaken on lamb carcass price and concentrate price, variables that have previously been shown to significantly impact net profit [10,37]. Due to the fact that the simulated system was a grass-based production system, where grazed grass and silage accounted for approximately 98% of the diet, the impact of adjusting concentrate prices had little impact on overall net profit, despite concentrate input being one of the main variable costs in Irish sheep production systems. Of late, prices for raw cereals and, subsequently, concentrate feeds have increased globally; therefore, the impact of concentrate price on overall net profit would be expected to be much greater, albeit at small inclusion rates. Alternatively, overall net profit was shown to be sensitive towards a shift in lamb price; as costs of production were unaffected, the proportion of the carcass value retained as net profit increased when lamb price increased. A large change in net profit (26.7% to 41.5%) was observed when lamb prices increased by 10% which can be explained by the fact that lamb price affected every lamb sold within the system, where, in comparison, concentrates were only fed to a small proportion of slower-growing lambs.

## 5. Conclusions

This simulation allowed the comparison of high-genetic-merit animals of New Zealand and Irish origin to take place, using animal performance parameters [4,5,6], and showed the performance of NZ animals to be superior to that of the Irish animals, both High and Low genetic merit, across a number of traits. Furthermore, although differences between High Irish and Low Irish animals were often small [4,5,6], this simulation demonstrated the impact the accumulation of a number of key performance indicators can have on overall farm profitability. Results from the current study showed that the number of lambs slaughtered was the main driver of total farm income within all three scenarios investigated. Potential exists to increase the number of lambs weaned/slaughtered per hectare through the selection of animals of high genetic merit, regardless of origin, over those of low genetic merit. Therefore, the Irish sheep industry should continue to use the €uro-star Replacement index when selecting animals for breeding, in order have a positive impact on farm performance and subsequent profitability.

## Figures and Tables

**Table 1 animals-13-02908-t001:** Model input assumptions across the three modelled scenarios: New Zealand (NZ), high genetic merit Irish (High Irish) and low genetic merit Irish (Low Irish).

Scenario	NZ	High Irish	Low Irish
Farm size (ha)	20	20	20
Ewes joined to ram	240	240	240
Stocking rate (ewes/ha)	12	12	12
Scanning rate (lambs/ewe)	1.84	1.62	1.59
Weaning rate (lambs/ewe joined)	1.46	1.28	1.26
Replacement rate (%)	28.80	34.14	35.59
Nitrogen use (kg/ha)	130	130	130

**Table 2 animals-13-02908-t002:** Comparison of physical details of the three modelled scenarios (New Zealand (NZ), high genetic merit Irish (High Irish) and low genetic merit Irish (Low Irish)) across animal numbers, flock performance, feed (grass and concentrate usage) and labour requirements.

Scenario	NZ	High Irish	Low Irish
Ewes joined to the ram ^a^	240	240	240
Lamb mortality (%) ^a^	16	16	16
Weaning weight ^a^	33.99	33.14	32.37
Lambs weaned	350	308	303
Lambs weaned/ha	17.5	15.4	15.2
Lambs slaughtered/ha	14.05	11.40	11.00
Total carcass sold (kg/ha)	282	230	222
Drafted by 1 October (%)	94	88	84
Total concentrates (kg/ha)	273	259	263
Grass grown (kg DM/ha) ^a^	10,792	10,792	10,792
Grass utilised (kg DM/ha) ^a^	9180	9180	9180
Total labour requirement (h)	1317	1304	1300
Total hired labour (h)	43	40	39

^a^ Model assumptions based on results from [4,5,6].

**Table 3 animals-13-02908-t003:** Trading profit and loss accounts for each scenario (New Zealand (NZ), high genetic merit Irish (High Irish) and low genetic merit Irish (Low Irish)), reported on a net profit per hectare basis (EUR/ha).

Scenario	NZ (EUR/ha)	High Irish (EUR/ha)	Low Irish (EUR/ha)
Wool	20	21	21
Lamb	1437	1175	1124
Culls	215	262	275
Surplus silage	0	6	11
Total farm receipts (A)	1672	1457	1419
Variable costs			
Concentrates	81	77	78
Straw	53	55	56
Fertilizer	221	221	221
Lime	20	20	20
Reseeding	50	50	50
Livestock purchases	27	27	27
Dead animal disposal	14	14	14
Machinery hire	32	33	33
Silage making	79	83	84
Veterinary and medicine	177	174	175
Carcass processing levies	15	13	13
Machinery operation and repair	94	94	94
Total variable costs (B)	863	862	866
Gross margin (A−B)	809	595	554
Fixed costs			
Car use	60	60	60
Electricity and phone	27	27	27
Hired labour	21	20	20
Farm insurance	40	40	40
Buildings depreciation	43	44	44
Machinery depreciation	49	49	49
Total fixed costs (C)	241	240	240
Total farm costs (B + C)	1104	1103	1106
Farm net profit per hectare (A−B−C)	514	299	258

**Table 4 animals-13-02908-t004:** A comparison of financial performance indicators for each scenario (New Zealand (NZ), high genetic merit Irish (High Irish) and low genetic merit Irish (Low Irish)).

Scenario	NZ	High Irish	Low Irish	
Number of lambs slaughtered	281	228	220	
Carcass value	EUR 102	EUR 103	EUR 102	
Total income per lamb	EUR 119	EUR 128	EUR 129	
Total concentrate costs per lamb	EUR 5.76	EUR 6.76	EUR 7.12	
Total cost of production per lamb	EUR 82.35	EUR 101.42	EUR 105.72	
Net profit per lamb	EUR 36.54	EUR 26.19	EUR 23.44	
Price received per kg carcass	EUR 5.10	EUR 5.11	EUR 5.06	
Cost of production per kg carcass	EUR 4.11	EUR 5.04	EUR 5.23	
Net profit per kg carcass	EUR 1.82	EUR 1.30	EUR 1.16	

**Table 5 animals-13-02908-t005:** The impact of fluctuations in lamb and concentrate price on total net profit of a 20-hectare farm for each scenario (New Zealand (NZ), high genetic merit Irish (High Irish) and low genetic merit Irish (Low Irish)).

	NZ		High Irish		Low Irish	
	Total Net Profit	% Change	Total Net Profit	% Change	Total Net Profit	% Change
Base	EUR 10,279		EUR 5982		EUR 5152	
10% change in lamb price	±EUR 2747	±26.7%	±EUR 2351	±39.3%	±EUR 2138	±41.5%
10% change in concentrate price	±EUR 162	±1.6%	±EUR 155	±2.6%	±EUR 157	±3.0%

## Data Availability

This research did not receive any specific grant from funding agencies in the public, commercial or not-for-profit sectors.

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
