# Peer review of "Investigating How Genetic Merit and Country of Origin Impact the Profitability of Grass-Based Sheep Production Systems"

_animals, 2023, doi:10.3390/ani13182908_

Round 1
Reviewer 1 Report
1) Methodology: how were the economic indices obtained? It is necessary to describe the methodology in detail.
2) Is it possible to draw these conclusions and discuss what was discussed with the number of animals that were used to carry out the analyses?
3) How were the selection indexes obtained? It has not been sufficiently described in the material and methods.
Moderate editing of English language required
Reviewer 2 Report
Overall comments
In the manuscript animals-2509805 Investigating how genetic merit and country of origin impact the profitability of grass-based sheep production systems, authors used a the bio-economic Teagasc Lamb production Model to simulate and assess the profitability of sheep production systems that varied in maternal genetic merit (Hig or Low) and country of origin (New Zealand or Ireland). The paper appears well written, although needs some important explication. Indeed, auhtors should clarly explain why they compared NZ ewes vs low and high IE maternal genetic merit ewes. They justified their chose only taking in account results achieved in previous researches, consequently the paper seems part of a more extended research and could be considered as short communication. Moreover, concerning the sensitive analysis conducted authors supposed a fluctation by ±10% of lamb and concentrate price without give any explication o the choice done in terms of production factors choose and in terms of concentrate price fluctuation choose, taking also in account that in the last years the price fluctuation of corn and soybean which are the main used cocnentrated used in livestock produciton, for example, has been higher in some cases. Moreover, authors should explain in exhaustive manner wht they used in sensitivity analysis the concentrate price when it incidence in dry matter intake was only 2%, instead the price of others input such as fertilisers costs, lbour costs and so on. Finally I suggest major revision of the paper to be accepted.
Round 2
Reviewer 1 Report
The author's answer clarified the purpose of the manuscript and the methodology used. The inclusions made in the text changes are sufficient.
Author Response
Animal & Grassland Research and Innovation Centre,
Teagasc,
Mellows Campus,
Athenry,
Co. Galway
Ireland
H65 R718
22/08/2023
Dear Sir / Madam,
Many thanks for taking the time to review my manuscript ID 2509805, entitled ‘Investigating how genetic merit and country of origin impact the profitability of sheep production systems’. Each of your comments have been addressed below and in the revised manuscript.
Reviewer 1
Comments and Suggestions for Authors:
The author's answer clarified the purpose of the manuscript and the methodology used. The inclusions made in the text changes are sufficient. Many thanks for taking the time to review the manuscript.
Yours sincerely,
Fiona McGovern
Reviewer 2 Report
Concerning the manuscript animals-2509805 authors answers in satisfying manner to the questions posed in the previous review step and the revised versions appears improved in a sufficient manner. Indeed, authors explained in satisfant manner in the new version the reasons to compare IRL animals with NZ animals. Regarding my request of clarification concerning the sensitive analysis conducted, authors answered in a satisfant manner to my questions although the fluctation by ±10% of lamb concentrate price appears still low considering the actual prices of cereals and protein sources at a global scale. Moreover, concerning the incidence of concentrate price variaitons in the diet, its amount (2% of total DMI) appears very low and consequently results obtained applicable only in areas with climate that permits long grazing seasons.
Author Response
Animal & Grassland Research and Innovation Centre,
Teagasc,
Mellows Campus,
Athenry,
Co. Galway
Ireland
H65 R718
22/08/2023
Dear Sir / Madam,
Many thanks for taking the time to review my manuscript ID 2509805, entitled ‘Investigating how genetic merit and country of origin impact the profitability of sheep production systems’. Each of your comments have been addressed below and in the revised manuscript.
Reviewer 2
Comments and Suggestions for Authors:
1) Concerning the manuscript animals-2509805 authors answers in satisfying manner to the questions posed in the previous review step and the revised versions appears improved in a sufficient manner. Indeed, authors explained in satisfant manner in the new version the reasons to compare IRL animals with NZ animals. Regarding my request of clarification concerning the sensitive analysis conducted, authors answered in a satisfant manner to my questions although the fluctation by ±10% of lamb concentrate price appears still low considering the actual prices of cereals and protein sources at a global scale. Moreover, concerning the incidence of concentrate price variations in the diet, its amount (2% of total DMI) appears very low and consequently results obtained applicable only in areas with climate that permits long grazing seasons.
Many thanks for taking the time to review the manuscript. The authors acknowledge that given the current situation a 10% fluctuation is on the lower side but can be increased by the reader manually to give implications for increments currently relevant. The manuscript has been updated to reflect this. Similarly the manuscript has been updated to reflect the fact that results obtained herein are from an area where a longer grazing season is permitted.
Yours sincerely,
Fiona McGovern